# Double Mutations in a Patient with Early-Onset Alzheimer’s Disease in Korea: An *APP* Val551Met and a *PSEN2* His169Asn

**DOI:** 10.3390/ijms24087446

**Published:** 2023-04-18

**Authors:** Heewon Bae, Kyu Hwan Shim, Jang Yoo, Young-Soon Yang, Seong Soo A. An, Min-Ju Kang

**Affiliations:** 1Department of Neurology, Veterans Medical Research Institute, Veterans Health Service Medical Center, Seoul 05368, Republic of Korea; hwbae0601@gmail.com; 2Department of Bionano Technology, Gachon University, Seongnam 13120, Republic of Korea; 3Department of Nuclear Medicine, Veterans Health Service Medical Center, Seoul 05368, Republic of Korea; 4Department of Neurology, Soonchunhyang University College of Medicine, Cheonan Hospital, Cheonan 31151, Republic of Korea

**Keywords:** Alzheimer’s disease, double mutations, amyloid precursor protein, presenilin 2

## Abstract

The etiology of early-onset Alzheimer’s disease (EOAD) is associated with alterations in the production of amyloid beta (Aβ) species caused by mutations in the *APP*, *PSEN1*, and *PSEN2* genes. Mutations affect intra- or inter-molecular interactions and processes between the γ-secretase complex and amyloid precursor protein (APP), leading to the aberrant sequential cleavage of Aβ species. A 64-year-old woman presented with progressive memory decline, mild right hippocampal atrophy, and a family history of Alzheimer’s dementia (AD). Whole exome sequencing was performed to evaluate AD-related gene mutations, which were verified by Sanger sequencing. A mutation-caused structural alteration of APP was predicted using in silico prediction programs. Two AD-related mutations, in *APP* (rs761339914; c.G1651A; p.V551M) and *PSEN2* (rs533813519; c.C505A; p.H169N), were identified. The *APP* Val551Met mutation in the E2 domain may influence APP homodimerization through changes in intramolecular interactions between adjacent amino acids, altering Aβ production. The second mutation was *PSEN2* His169Asn mutation, which was previously reported in five EOAD patients from Korea and China, with a relatively high frequency in the East Asian population. According to a previous report, the presenilin 2 protein was predicted to result in a major helical torsion by *PSEN2* His169Asn mutation. Notably, the co-existence of *APP* Val551Met and *PSEN2* His169Asn may induce a synergistic effect by both mutations. Future functional studies are needed to clarify the pathological effects of these double mutations.

## 1. Introduction

Mutations in the genes encoding amyloid precursor protein (APP), presenilin 1 (PS1), and presenilin 2 (PS2) proteins affect amyloid beta (Aβ) generation, leading to altered APP cleavage and the pathophysiology underlying Alzheimer’s disease (AD) progression [1]. Patients with early-onset Alzheimer’s disease (EOAD) have numerous mutations in the aforementioned genes related to the deposition of extracellular Aβ plaques and intraneuronal neurofibrillary tangles by tau proteins in the brain [2]. Aβ peptides are generated by the sequential cleavage of APP by β-secretase and γ-secretase [3]. In the first stage by β-secretase cleavage, the ectodomain of APP is removed, leaving a 99 amino acid C-terminal fragment (C99 or APP CTFβ) on the membrane [4]. In a previous report, a Swedish double mutation located adjacent to the β-secretase cleavage site of APP has been found to increase the production and secretion of Aβ [5]. C99 is cleaved from the transmembrane domain by the γ-secretase complex to release the APP intracellular domain (AICD) from the membrane to the cytoplasm, followed by sequential cleavage to produce Aβ peptides of different lengths [6]. PS1 and PS2 are crucial γ-secretase components, which belong to a multiprotein complex composed of intramembrane proteases as well as nicastrin, presenilin enhancer 2, and anterior pharynx defective 1 [7]. Mutations in *PSEN1* or *PSEN2* may cause the structural instability of γ-secretase, which affects the production of longer Aβ peptides during APP processing [8]. Based on the Alzforum database (http://www.alzforum.org/mutations, assessed on 14 February 2023), mutations in *PSEN1* have been reported in 357 cases, which typically resulted in increased Aβ_42_ secretion compared with Aβ_40_ secretion [9]. *PSEN2*, which has 87 reported rare mutations, is highly homologous to *PSEN1*. Recent studies have reported several *PSEN2* mutations in Asia, mostly in Korea and China [10]. Disease penetrations of *PSEN2* mutations in AD seemed to be variable with clinical profiles and the age of onset ranging from 40 to 80 years across patients [11].

This article describes the clinical phenotype of a 64-year-old Korean woman with EOAD who had double mutations in *APP* and *PSEN2*, which included a novel *APP* Val551Met mutation and a *PSEN2* His169Asn previously reported in Asian patients [12,13,14,15].

## 2. Results

### 2.1. Clinical Characterization

A 63-year-old right-handed woman with 12 years of education presented with progressive memory decline for 2 years before the first hospital visit. She had symptoms of forgetting passwords or being unable to recall a person’s name. She regularly misplaced items and often forgot words or stories while speaking. However, she managed her household without problems. She scored 27/30 on the Mini-Mental Status Examination; moreover, she had a global deterioration scale (GDS) score of 2. Neuropsychological testing revealed visuospatial dysfunction. Brain magnetic resonance imaging revealed mild right hippocampal atrophy (Figure 1A). Her initial diagnosis was AD-caused mild cognitive impairment. After 2 years, she presented with deterioration of memory deficits. She could not remember what she had just said; moreover, she could not find a familiar route. She scored Mini-Mental Status Examination score and GDS score of 24/30 and 3, respectively. The acetylcholinesterase inhibitor, donepezil, was administered at a dosage of 10 mg/day.

A family history could not be fully defined; however, we could not rule out a possible familial case of the disease. Her mother had been diagnosed with Alzheimer’s disease in her early 70s and died in her 80s. There was no available information on grandparents or siblings. The patient had three siblings; additionally, all siblings lacked any neurodegenerative symptoms and refused genetic testing (Figure 1B).

### 2.2. Genetic Analysis and In Silico Prediction

Mutations were found in the *APP* (rs761339914) and *PSEN2* (rs533813519) genes. Heterozygous substitutions of *APP* (G to A) and *PSEN2* (C to A) were verified through standard Sanger sequencing, which altered the amino acids of APP at codon 551 (Valine to Methionine) and PS2 at codon 169 (Histidine to Asparagine) (Figure 1C,D). In the TOPMED and ExAC databases, the *APP* rs761339914 mutation was rarely reported, with frequencies of 0.000008 and 0.000025, respectively. The frequencies of *PSEN2* rs533813519 were 0.000165 in ExAC and 0.000057 in TOPMED. Contrastingly, in the database of the Korean population, the *PSEN2* mutation frequency was relatively higher (0.00078 in 8.3 KJPN and 0.0041 in KOREAN).

The *APP* Val551Met mutation was likely damaging and tolerated in Polyphen2 and SIFT, respectively. To investigate the effects of an amino acid substitution in mutant APP, 3D structures of the mutated protein sequences were predicted and visualized using Discovery Studio software. There were no differences in the structures of *APP* Val551Met and wild-type APP (Figure 1E,F). Contrastingly, in the intra-interaction among amino acids, Met551 exhibited a hydrophobic interaction with Leu480 of the other α-helix chain rather than Ala553. In silico analysis of *PSEN2* His169Asn mutation has been previously described [12]. The His169Asn mutation was predicted to induce major helix torsion and additional stress on the surface interacting with transmembrane domain III.

Notably, cells with Tyr538His and Val562Ile mutations near Val551Met have shown decreased Aβ_40_ production [16]. Each mutation had different trends on Aβ processing and secretion, but this is inferred to be due to the intrinsic properties of the mutations. Among the mutations in the APP E2 domain, the Arg486Trp mutation was described in one patient with familial EOAD from a British cohort [17]. In other de novo cases, Lys496Gln has been reported in EOAD while Tyr538His and Val562Ile have been reported in late-onset AD [18,19,20]. Mutations in the E2 domain are considered a low-risk factor contributing to AD etiology and cannot solely cause the disease (Table 1).

## 3. Discussion

Genetic analysis of rare variants in patients with EOAD may help elucidate the pathogenesis underlying AD by identifying protein structural or functional changes. In this study, a Korean patient with EOAD presented with a novel *APP* Val551Met mutation and a sixth *PSEN2* His169Asn mutation in East Asia.

The *APP* Val551Met mutation was located at the ectodomain (E2), outside of the cleavage site by β- and γ-secretases. This novel mutation has a very low frequency in other genome-wide association databases (TOPMED and ExAC). The pathogenicity of mutations outside the β-secretase cleavage site remains unclear. However, mutations in the E2 could influence APP processing dynamics as the E2 domain of APP is responsible for APP self-dimerization [23,24,25]. Substitution mutations of the E2 domain could influence APP homodimerization, resulting in the alteration of Aβ production. For example, Aβ production is increased 6–8 times upon the formation of a stable APP dimer through the expression of a cysteine mutation in the ectodomain juxtamembrane region [24]. Therefore, the reduced APP dimerization resulting from the mutation in the ectodomain could have opposite effects. Recent studies have focused on predicting disease susceptibility by developing a polygenic risk score (PRS) for each disease-related variant [26]. Clinical implementation of PRS may facilitate diagnosis or inform treatment choices at an early stage of the disease. Regarding the application of the PRS strategy to rare variants, Lali et al. proposed a predictive rare variant genetic risk score to delineate the effect of the risk variant burden in numerous genes [27]. Although one *APP* Val551Met may have minor effects, its combination with *PSEN2* His169Asn could facilitate the onset of AD.

*PSEN2* His169Asn has been reported in one and four cases in Korea and China, respectively (Table 2) [12,13,14,15]. The clinical manifestation of the previous cases was presented to an early onset age of less than 65 years, with a progressive memory impairment similar to our patient. Although all cases strongly demonstrate the high-penetrance gene mutations of the AD pathology, the contribution of the mutation to the AD pathophysiology could be relatively modest given the reported age of onset in these patients compared with that in patients with *PSEN1* mutations. *PSEN2* His169Asn is located in TMIII, which is a highly conserved domain in proteins that includes other AD-related variants, including Met174Ile, Met174Val, Ser175Cys, Ser175Phe, and Phe183Ser. *PSEN2* His169Asn was predicted to be structurally pathogenic by causing a significant twist in the helix, resulting from the substitution of histidine for asparagine [12]. Notably, since this mutation was only found in the Asian population, it could be a founder’s mutation continuously monitored in East Asia.

A limitation of this study is that the family members of the patient did not undergo genetic testing for determining the origin of the *APP* Val551Met and *PSEN2* His169Asn mutations.

## 4. Materials and Methods

Consent was obtained for processing genetic and clinical data for research purposes. Likely AD diagnosis was based on the criteria of the National Institute on Aging-Alzheimer’s Association. This study was approved by the institutional review board of the Veterans Healthcare Medical Center (2021-06-015).

### 4.1. DNA Purification and Genetic Analysis

Whole blood was collected in EDTA tubes and stored at −20 °C until use. Purified genomic DNA was collected from the whole blood using the QIAamp DNA Blood Maxi Kit (Qiagen, Hilden, Germany), following the manufacturer’s instructions. The quantity and quality of the purified DNA were analyzed, with samples being kept at −20 °C for further analysis. Whole-exome sequencing was applied to identify genetic variants in genes associated with neurodegenerative disorders. Purified DNA samples were sequenced using the Illumina platform. To confirm the presence of mutations, primers suitable for exons in *APP* and *PSEN2* were designed (*APP*, forward primer: 5′-CCTTCTCGTTCCTGACAAGTGC-3′ and reverse primer: 5′-GGCAGCAACATGCCGTAGTCAT-3′; *PSEN2*, forward primer: 5′-CCCACCGTGGATGACCTAATA-3′ and reverse primer: 5′-CACTTGTGCCACATGGAGAG-3′), followed by PCR-based gene analysis using typical Sanger sequencing (Bioneer Inc., Daejeon, Republic of Korea). BigDye Terminator Cyclic sequencing was performed using an ABI 3730XL DNA Analyzer (Applied Biosystems, Foster, CA, USA). The NCBI Gene (http://www.ncbi.nlm.nih.gov/gene, assessed on 8 April 2022) and UniProt (http://www.uniprot.org, assessed on 8 April 2022) databases were used to verify gene and protein sequences.

### 4.2. In Silico Analyses

Two screening tools, PolyPhen-2 (http://genetics.bwh.harvard.edu/pph2/, assessed on 8 April 2022) and SIFT (https://sift.bii.a-star.edu.sg/index.html, assessed on 8 April 2022) were used to determine the nature of the mutations, classifying them as either benign or potentially damaging. The possible pathogenicity of the APP and PS2 proteins was investigated using mutation-induced structural alterations. The 3D protein structures of the normal and mutant APP and PS2 proteins were estimated using Missense3D (http://missense3d.bc.ic.ac.uk/missense3d/, assessed on 17 October 2022). Visualization and interaction with surrounding amino acids were performed using Discovery Studio 3.5 Visualizer (BIOVIA, San Diego, CA, USA).

## 5. Conclusions

In this study, a Korean patient was presented with double mutations in both *APP* and *PSEN2*. To the best of our knowledge, this is the first case with double *APP* and *PSEN2* rare variants. Further studies are warranted to investigate the possibility of independent synergistic effects of *APP* Val551Met and *PSEN2* His169Asn. Similar to PRS, which recently incorporated common variants to predict disease susceptibility, integrating rare low-penetrance variations, including the double mutations, could help elucidate the mutations implicated in AD. Specifically, a genetic screening panel for East Asian-specific rare variants should include *PSEN2* His169Asn to improve diagnostic accuracy and help elucidate AD pathogenicity.

## Figures and Tables

**Figure 1 ijms-24-07446-f001:**
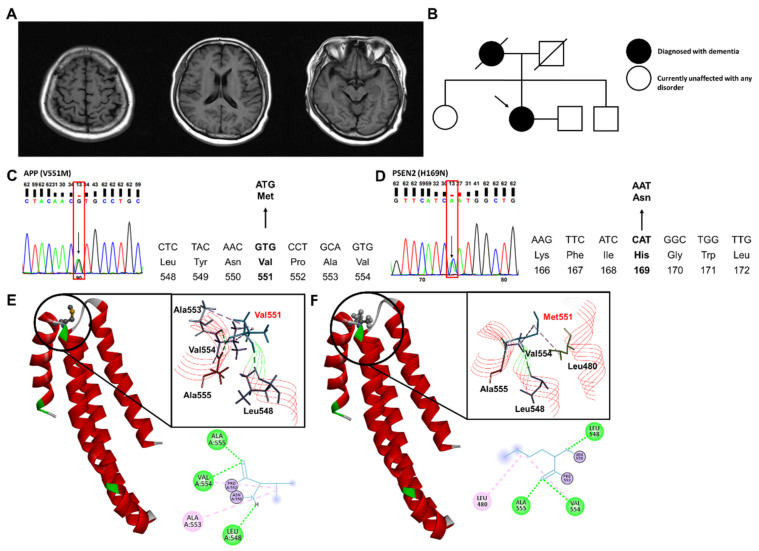
Brain imaging, family pedigree, and genetic analysis of a patient as well as prediction of the mutation-induced protein structure. (**A**) Axial FLAIR, coronal T2 images of brain MRI, with the arrows indicating the right hippocampal atrophy. (**B**) Family tree identifying the proband (arrow) with the *APP* and *PSEN2* mutations. (**C**,**D**) Sanger sequencing of *APP* and *PSEN2* variant. Heterozygous variants were indicated by a black arrow and red box. (**E**,**F**) The prediction of the mutation-induced alterations in the intramolecular interactions. 2D and 3D prediction of the structure and interactions between amino acids in the wild-type (**E**) and mutant APP (**F**). Green and pink dot lines represent hydrogen bond and alkyl interaction, respectively.

**Table 1 ijms-24-07446-t001:** Mutations in the E2 domain of APP.

Gene	Mutation	Family History	Functional Data	Clinical Phenotype	Ref.
*APP*	E380K	De novo	Likely damaging in PolyPhen-2 and SIFT	LOAD	[18]
	R468H	De novo	No changes in Aβ_40_ or Aβ_42_ production in cells	None	[21]
	A479S	De novo		None	[22]
	R486W	Familial	Likely damaging by in silico algorithms	EOAD	[17]
	K496Q	De novo	Slight increase in Aβ_40_ in cells	EOAD	[19]
	A500T	De novo	No changes in Aβ_40_ or Aβ_42_ production in cells	-	[21]
	Y538H	De novo	Decrease in Aβ_40_ and Aβ_42_ in cells	LOAD	[20]
	V562I	-	Slight decrease in Aβ_40_ secretion	-	[20]

**Table 2 ijms-24-07446-t002:** List of previously reported patients with *PSEN2* His169Asn mutations.

Country	Sex	APOE Type	Family History	Age of Onset	First Clinical Symptom	Brain Imaging	Clinical Phenotype	Ref.
Korea	Female	ε3/3	De novo	50	Memory loss	Atrophy and hypometabolism in the left temporal lobe	EOAD	[12]
China	Female	ε3/4	Familial	68	Memory loss	Bilateral temporal lobe atrophy; hypometabolism; positive PIB PET	LOAD	[13]
	Male	ε3/3	De novo	62	Language difficulty and behavioral changes	Mild atrophy; hypometabolism; negative PIB PET	Frontotemporal dementia	[13]
	-	ε3/3	-	64	-	-	EOAD	[14]
	Male	-	Familial	63	Memory loss	Temporal and bilateral hippocampal atrophy	EOAD	[15]

## Data Availability

The data presented in this study are available upon request from the corresponding author.

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
