# Peer review of "Double Mutations in a Patient with Early-Onset Alzheimer’s Disease in Korea: An APP Val551Met and a PSEN2 His169Asn"

_ijms, 2023, doi:10.3390/ijms24087446_

Round 1

Reviewer 1 Report

The manuscript presented for review is very interesting and fits into the trend of searching for the pathogenesis of Alzheimer's disease, which is extremely important because it is one of the most common causes of dementia worldwide. Learning the full pathogenesis of its development, especially in the face of aging societies in the world, is extremely important because it can contribute to earlier diagnosis and be a target for new therapy. The results obtained in the work are interesting and can be a starting point for further research not only by the authors themselves, but also by the entire scientific community. The work is written clearly, but in the opinion of the reviewer, it requires a few minor changes:

- names of genes within the entire manuscript should be written in italics

- no dates of access to databases (please complete for all databases or in silico tools used in the work)

- no primer sequences in Materials and Methods

- page 2 verse 92-93 "The APP Val551Met mutation was probably damaging and tolerated in Polyphen2 92 and SIFT, respectively." - if the authors performed the analysis themselves, they should include the results from these tools in the form of a table or figures in the paper; and should also describe them in the section Materials and Methods - in silico analysis;

- Table 1 - in the opinion of the reviewer, Table 1 and its description should be included in the Results section

- Table 1 - whether the authors prepared this list on the basis of their own research or  references. If based on the references, there are no citations in the Table

Author Response

Dear Editor and Reviewers,

We greatly appreciate your thoughtful comments that helped us to improve the manuscript's quality and readability. It is pleasure to receive positive feedback from all the reviewers. Upon reviewing our revisions, we hope that you will find the manuscript acceptable for publication in International Journal of Molecular Sciences.

Response to Reviewer 1 Comments

The manuscript presented for review is very interesting and fits into the trend of searching for the pathogenesis of Alzheimer's disease, which is extremely important because it is one of the most common causes of dementia worldwide. Learning the full pathogenesis of its development, especially in the face of aging societies in the world, is extremely important because it can contribute to earlier diagnosis and be a target for new therapy. The results obtained in the work are interesting and can be a starting point for further research not only by the authors themselves, but also by the entire scientific community. The work is written clearly, but in the opinion of the reviewer, it requires a few minor changes:

Point 1: names of genes within the entire manuscript should be written in italics

Response 1: Thanks for your correction. We have revised all names of genes.

Point 2: no dates of access to databases (please complete for all databases or in silico tools used in the work)

Response 2: As reviewer’s comment, we have stated the dates of access of databases. (p.#6, lines #18-25)

Point 3: no primer sequences in Materials and Methods

Response 3: As reviewer’s comment, we have added the sequences of primers. (p.#6, lines #8-11)

Point 4: page 2 verse 92-93 "The APP Val551Met mutation was probably damaging and tolerated in Polyphen2 92 and SIFT, respectively." - if the authors performed the analysis themselves, they should include the results from these tools in the form of a table or figures in the paper; and should also describe them in the section Materials and Methods - in silico analysis;

Response 4: PolyPhen-2 provides results in the categories of benign and potentially damaging, and SIFT reports tolerated or not tolerated outcomes in a simplified format. The website accessed for the predictions was specified in the Methods section. (p.#2-3, 2. Results, paragraph 4)

Point 5: - Table 1 - in the opinion of the reviewer, Table 1 and its description should be included in the Results section

- Table 1 - whether the authors prepared this list on the basis of their own research or  references. If based on the references, there are no citations in the Table

Response 5: Thank you for your opinion. Table 1 was moved to the Results section, and all references were stated.

Reviewer 2 Report

Before reviewing the actual work, I would like to know if the authors are are aware of the following article: https://doi.org/10.1126/science.add9993.

Considering the serious implications of what the article claims and also because of Science.org itself opened an investigation on the matter, a critical view of the subject of the paper could be of interest.

Beyond the cited publication itself, there are many important voices that rise questions. I cite here from Alzheimer's Research UK:

"Prof Sir John Hardy from UCL developed the amyloid hypothesis in the 1990s, and told us he has long had doubts about the paper central to today’s story. “I have never thought this paper was important, and I don’t think I have ever referred to it in my own work,” he says. In fact, Hardy is sceptical about the role of amyloid oligomers in Alzheimer’s. “I don’t think their toxicity has ever been satisfactorily demonstrated,” compared to other forms of amyloid." 

I am sorry for not providing a typical review, but I think the very foundation of some theories needs sometimes a (brief) discussion, before deepening the research on established grounds.

Author Response

Dear Editor and Reviewers,

We greatly appreciate your thoughtful comments that helped us to improve the manuscript's quality and readability. It is pleasure to receive positive feedback from all the reviewers. Upon reviewing our revisions, we hope that you will find the manuscript acceptable for publication in International Journal of Molecular Sciences.

Response to Reviewer 2 Comments

Point 1: Before reviewing the actual work, I would like to know if the authors are are aware of the following article: https://doi.org/10.1126/science.add9993.

Considering the serious implications of what the article claims and also because of Science.org itself opened an investigation on the matter, a critical view of the subject of the paper could be of interest.

Beyond the cited publication itself, there are many important voices that rise questions. I cite here from Alzheimer's Research UK:

"Prof Sir John Hardy from UCL developed the amyloid hypothesis in the 1990s, and told us he has long had doubts about the paper central to today’s story. “I have never thought this paper was important, and I don’t think I have ever referred to it in my own work,” he says. In fact, Hardy is sceptical about the role of amyloid oligomers in Alzheimer’s. “I don’t think their toxicity has ever been satisfactorily demonstrated,” compared to other forms of amyloid."

I am sorry for not providing a typical review, but I think the very foundation of some theories needs sometimes a (brief) discussion, before deepening the research on established grounds.

Response 1: Thank you for your comment regarding the recent article published in Science and its implications on the amyloid hypothesis in Alzheimer's disease. We are aware of the article and the ongoing investigation of the claims. While we acknowledge the importance of considering critical views and discussing the foundation of theories, we believe that the amyloid hypothesis remains a valid and valuable approach in understanding Alzheimer's disease pathology.

As you mentioned, Professor Sir John Hardy from UCL, who developed the amyloid hypothesis in the 1990s, has expressed skepticism about the role of amyloid oligomers in Alzheimer's. However, it is important to note that other researchers have provided evidence supporting the amyloid hypothesis and the role of amyloid oligomers in Alzheimer's disease. In fact, many ongoing clinical trials are still based on this hypothesis and are targeting amyloid to prevent or slow down the progression of the disease. Lecanemab (Leqembi™), shown to bind to both soluble and insoluble forms of beta-amyloid, has received accelerated approval as a treatment for early Alzheimer's from the U.S. Food and Drug Administration (FDA).

Therefore, we believe that it is important to continue investigating the amyloid hypothesis and related pathways, such as tau and inflammation, to fully understand the complex mechanisms underlying Alzheimer's disease.
